# Recent Developments on Semiconducting Polymer Nanoparticles as Smart Photo-Therapeutic Agents for Cancer Treatments—A Review

**DOI:** 10.3390/polym13060981

**Published:** 2021-03-23

**Authors:** N. Sanoj Rejinold, Goeun Choi, Jin-Ho Choy

**Affiliations:** 1Intelligent Nanohybrid Materials Laboratory (INML), Institute of Tissue Regeneration Engineering (ITREN), Dankook University, Cheonan 31116, Korea; sanojrejinold@dankook.ac.kr (N.S.R.); goeun.choi@dankook.ac.kr (G.C.); 2College of Science and Technology, Dankook University, Cheonan 31116, Korea; 3Department of Nanobiomedical Science and BK21 PLUS NBM Global Research Center for Regenerative Medicine, Dankook University, Cheonan 31116, Korea; 4Department of Pre-medical Course, College of Medicine, Dankook University, Cheonan 31116, Korea; 5Tokyo Tech World Research Hub Initiative (WRHI), Institute of Innovative Research, Tokyo Institute of Technology, Yokohama 226-8503, Japan

**Keywords:** semiconducting polymers, photo-therapy, applications, limitations, future scope

## Abstract

Semiconducting polymer nanoparticles (SPN) have been emerging as novel functional nano materials for phototherapy which includes PTT (photo-thermal therapy), PDT (photodynamic therapy), and their combination. Therefore, it is important to look into their recent developments and further explorations specifically in cancer treatment. Therefore, the present review describes novel semiconducting polymers at the nanoscale, along with their applications and limitations with a specific emphasis on future perspectives. Special focus is given on emerging and trending semiconducting polymeric nanoparticles in this review based on the research findings that have been published mostly within the last five years.

## 1. Introduction

Cancer has been one of the deadliest diseases that human society has ever witnessed [1,2,3,4,5,6,7,8,9,10,11]. Though there have been dramatic improvements in early diagnosis, detection, and therapy, it still remains as deadly as ever before. With the advanced developments in cancer nanomedicine, it is now possible to combat cancers if detected early [12]. Nanomedicine [13,14,15,16], particularly use both fundamental science and bio-molecular engineering, in order to have better precision medicines [17]. Even though there have been various treatment regimens using nanomedicines, phototherapy is of great importance as it is minimally invasive and causes no side effects on the body [18,19,20,21,22]. There are two main streams of phototherapy, namely photo-thermal therapy (PTT) and photo-dynamic therapy (PDT), and both of them use light as their main source of action.

Typically, PTT works by absorbing infra-red (NIR) light (by rationally engineered photo absorbing agents) and converting it into therapeutic heat [23]. On the other hand, PDT use light energy to generate reactive oxygen species (ROS), particularly singlet oxygen (^1^O_2_), that can effectively hinder pathogenesis [24]. These two mechanisms can be combined to have multi-effect on better therapeutic outcome. A variety of nanomaterials such as organic and inorganic NIR sensitive agents have been explored for photo-therapy. Among them, semiconducting polymer nanoparticles (SPN) became very attractive recently due to their opto-electronic characteristics [25,26,27,28,29,30,31,32,33,34,35,36].

SPN have been used for photo-therapy including PTT, PDT and their combination owing to its aromatic structure with π-π interactions generating heat when exposed under NIR light [37,38]. Various semiconducting polymers (SPs) and conjugated polymers (CPs) have been identified, including polyaniline (PAN), polypyrrole (PPy), polydopamine (PDA) etc. [39,40,41]. The most important thing is that such SPs can easily form self-assembled structures at the nanoscale level, forming different types of SPN [42] (Table 1). Since SPN are considered as novel class of organic optical nanosystems, they have various benefits having brilliant opto-electrical properties with an excellent photostability, easy modification properties with a good biocompatibility for various biomedical applications including drug delivery[43] and bio-imaging [44].

SPN comprise mainly of hydrophobic but opto-electronically active SPs and amphiphilic polymer matrices [56]. The photo characteristics of SPN are mainly determined by the molecular structures of SPs. Such specific characteristics enable SPN to have relatively higher absorption coefficients and photo-stability compared to small molecular dyes and their self-assemblies [57]. Additionally, the organic and biologically inert nature of SPN reduces the risk of potential toxicity to living organisms, making it a perfect theranostic nanoagent for various photo-therapeutic strategies, including PTT, PDT, and their combined therapeutic strategy.

Since these SPN based nanomaterials are a new class of functional nanoparticles (NPs), it is important to have an overall insight into what has been going on in very recent years. The application of such NPs in imaging guided photo-therapy has to be reviewed. Therefore, we focus on reviewing details on recent development about design, synthesis, and application of SPNs for anti-cancer research area. 

Various novel polymeric structures have been designed through chemical functionalization via nanotechnology, which can improve its overall performance for phototherapy. In addition, such novel functionalized SPN have tunable optoelectronic properties improving photo-stability in terms of their fluorescence, chemiluminescence etc. In addition, such modified NPs have improved biocompatibility, and imaging capabilities either via photo-acoustic or NIR fluorescence imaging in in-vivo animal models [29,31,58,59,60,61].

Previously there were few similar review articles published, particularly for photo acoustic imaging (PAI) [62] by Zhou et al. (2018), whereas a review article by Chan et al. (2015) focused on “semiconducting polymer nanoparticles as fluorescent probes for biological imaging and sensing” [63]. Pu et al. (2016) also published a similar review article on recent advances in semiconducting polymer nanoparticles in in-vivo molecular imaging [64], Even though these review articles are not exactly similar to what we are trying to address, the one published in 2018 by Li et al. is somewhat similar to the scope of our current review. Li et al. combined applications of SPN on imaging methods and phototherapy [65]. Having said that, our current review article will be focusing on SPN based emerging NPs only on the phototherapy. In particular, emerging and trending SPN systems in the very recent research papers and those published in the last five years will be discussed for their wide range of applications.

Thus, the present review will be focused on design, applications, and future scope of such SPN in detail. We believe that this review would enlighten readers to understand the advantages and limitations of SPN research so that they can come up with improved scientific solutions to solve such issues. We also highlight SPN and its future perspectives in the current review article.

## 2. Semiconducting Polymer Nanoparticle (SPN)—How Do They Work? 

There have been wide varieties of SPs reported in the literature (Figure 1). The whole idea of such chemical modification seems to come up with either novel structures or enhancing the photostability as well as targetability to cancer cells. Most of the organic polymers are basically insulators. These kinds of organic polymers become conductors only when they have π-conjugated structures, in which variety of mechanisms exist, such as hopping, tunneling etc., aiding a smooth electron motion along the polymer backbone via overlaps in π-electron clouds. Generally, such π-electronic polymer systems in their pristine states are wide-band-gap semiconductors, also known as semiconducting polymers. As per discoveries in the 1970s, organic conjugated polymers and oligomers were found to have metallic traits upon heavy doping [66], as a term originally taken from inorganic semiconductor chemistry. The fact is that the conjugated polymer has a doping mechanism through oxidation (p-doping) or reduction (n-doping), respectively [67]. On the other hand, SPN based drug delivery systems became attractive mainly due to its opto-electronic properties in particular for various photo-thermal applications for anticancer therapy. In addition, they have tunable optoelectronic characteristics of metals or semiconductors and can still bear their innate mechanical characteristics and ease of preparation/manufacturing benefits, being polymers [68]. 

Even though there have been various SPN for bio-imaging and drug delivery applications [69], little is known for the applications in PTT, PDT, combined PTT/PDT, photo-immuno and photo-radio therapy. Thus, it is very important to have an overall look at their recent developments.

## 3. Design and Formulation Strategies of SPN 

It is of great importance to understand the major synthetic strategies of SPN based NPs to modify their drawbacks, so that one can come up with better formulation strategies. Most of the SPN based theranostic NPs were made using nano-precipitation, co-precipitation, emulsification and self-assembly methods. 

For example, NIR absorbing diketopyrrolopyrrole polymer P(AcIIDDPP) based NPs (DPP-IID-FA) were made via an emulsion method [45]. Similarly, thieno-isoindigo derivative-based donor acceptor (D-A) polymer (PBTPBF-BT) NPs have been made for NIR-II bio window for PTT against MDA-MB-231 xenograft mouse model [70]. On the other hand, the nanoprecipitation approach was used for thiadiazoloquinoxaline-based SPN for simultaneous imaging guided phototherapy by PAI/PTT for glioblastoma under NIR-II light range [46]. The same technique was adapted for two NIR absorbing molecules such as poly(cyclopentadithiophene-*alt*-benzothiadiazole) (SP1) and poly(acenaphthothienopyrazine-*alt*-benzodithiophene) (SP2) to prepare SPN [69].

In another work, the architecture of the organic SPNs+ has triple sections including a hydrophobic SP core, an anionic interlayer, and a cationic shell (+). The SP core acts as the PAI agent under NIR-I or NIR-II laser excitation, which was encapsulated by amphiphilic poly (styrene maleic anhydride) (PSMA) to obtain water-dispersed NPs (OSPNs−) with an anionic surface for further modification. Poly(l-lysine) (PLL) was subsequently adsorbed onto the OSPNs− surface via electrostatic interactions to enhance the cellular uptake. These SPN was made by nanoprecipitation method [71].

D–A CPs containing dibenzothiophene-S,S-dioxide based NPs have been developed using the nanoprecipitation method as effective photosensitizers. The obtained PTA5 copolymer had bright green emission and high photoluminescence quantum yields via the intercrossed excited state of local existed and charge transfer states. The PTA5 NPs were developed by loading them into a biocompatible polymer matrix [47]. In another study, immuno/phototherapeutic nano adjuvants were made using a hydration–sonication method [72].

Detailed information of synthetic methods for various SPN is shown in Figure 2. Typically, nano-precipitation has been found to be ideal for making synthetic polymer NPs [73] and can be extended for SPN formulation as well. This is mainly due to the following reasons such as ease of making, separation and high yield. However, in most of the cases, detailed mechanisms of such NPs were missing, and therefore, careful attention is required. What exactly happens in such formulation methods should be thoroughly understood as there will be always something interesting as we move from material to material, modifying their functionalities.

## 4. SPN in Photo-Therapy

Phototherapy, especially in treating cancers has been emerging since it can have minimal side effects compared to the existing conventional treatment modalities [77]. Cancer phototherapy approaches employ three main strategies: (1) PTT and (2) PDT and (3) Combined PTT/PDT. PTT or PDT or together can be combined with other therapies such as radiotherapy, immunotherapy etc. [23]. Therefore, it is important to have an overall view at each of these individual research areas in detail.

### 4.1. PTT

Theranostic NPs which can offer bright fluorescence and imaging capability with prominent photostability under laser irradiation is having great scope in terms of clinical applications. Especially NIR-II Window (1000–1700 nm) responsive SPN are far advantageous as they can be used for treating deep routed tumors easily, thanks to their low toxicity and high penetration capability of non-ionizing NIR-II waves [78]. However in reality, it is still challenging to develop such stable and ultra-small theranostic NPs for multi-purpose applications. 

A molecular engineering strategy has been utilized to make NIR-II emitting squaraine dyes. Initially, NIR-I squaraine dye SQ2 was made via ethyl-grafted 1,8-naphtholactam as donor units and square acid as acceptor unit in a donor–acceptor–donor (D–A–D) structure. A strong electron-withdrawing like Malonitrile, was added to enable square acid acceptor stronger, thereby shifting fluorescence towards NIR-II window. To translate NIR-II fluorophores SQ1 into effective theranostic agents, fibronectin-targeting SQ1 nano-probe consisting of 1,2-distearoyl-sn-glycero-3-phosphoethanolamine-*N*-[methoxy(polyethylene glycol)-2000 (DSPE-PEG2000) and pentapeptide CREKA was functionalized over these NPs. These nanoprobes showed high NIR-II imaging capability as confirmed on aa tumor model. Furthermore, the SQ1 nanoprobe could be used for PAI and PTT of tumors and was confirmed using 4T1 tumor bearing mouse model (Figure 3) [74].

PTT is one of the best non-invasive cancer treatments and a highly effective PTT always need safe and excellent PTT agents. To achieve this, Li et al. (2016) and team developed a 30 nm sized, highly photostable Pdots having 65% PTCE. The Pdots contain CPs and DPP with various thiophene derivatives (monothiophene, thienothiophene, bithiophene, and benzodithiophene) synthesized by a Stille coupling reaction. A nanoprecipitation technique was utilized to make 30 nm sized CPs NPs. According to the in-vitro and in-vivo analysis using a 4T1 tumor bearing mouse model, it was found that they were potential candidate for photo-thermal applications [48].

One of the major limitations associated with the PTT theranostic agents is poor stability and photothermal conversion efficiency (PTCE) and this could limit their clinical applicability. Novel CPs based Pdots have been reported to realize PAI-guided PTT. The Pdots were comprised of 4,8-bis[5-(2-ethylhexyl)thiophen-2-yl]-2,6-bis(trimethylstannyl)benzo[1,2-b:4,5-b′]dithiophene-6,6′-dibromo-*N*,*N*′-(2-ethylhexyl)isoindigo (BDT-IID) and were made via nanoprecipitation technique. These highly non-toxic stable Pdots had high PTCE ~ 45%. Figure 4 shows their PTT properties. As expected, the developed Pdots had improved PAI along with high PTT on MCF-7 tumor bearing mouse in their in-vivo analysis. These unique properties make them a suitable candidate for future clinical applications [49].

The “easy to make” versatile, economic and theranostic nanomaterials having imaging and therapeutic properties are of great interest in drug-delivery and imaging. To achieve such properties, a Gd^3+^-chelated poly(isobutylene-*alt*-maleic anhydride) (PMA) framework pendent with perylene-3,4,9,10-tetracarboxylic diimide (PDI) derivatives and PEG as effective theranostic nanostructure was developed for bi-modal PAI and MRI-guided PTT. Thus, the developed NPs chelated with Gd^3+^ (PMA–PDI–PEG–Gd NPs) showed a high T1 relaxivity coefficient (13.95 mM^–1^ s^–1^), even at the higher magnetic fields. Intravenously injected PMA–PDI–PEG–Gd NPs in Hela tumor bearing mouse model showed EPR assisted accumulation post injection of 3.5 h, as confirmed through PAI and MRI [50].

Customized medicines are always preferred as ideal theranostic agents. However, combined PTT and PAI agents with high biodegradation capability are rare. Lyu et al. (2018) studied such biocompatible and biodegradable SPNs having very high PAI capability along with improved PTT efficacy. These SPN were made of enzyme responsive vinylene bonds and can readily be transformed into a water-soluble matrix. Such vinylene backbone enhanced the PTCE to a great extent (~3-fold). These SPN based theranostic NPs showed better PAI and PTT on a 4T1 bearing tumor model [53].

A novel SPN (L1057 NPs) for NIR-II fluorescence imaging and PTT was made using nano-precipitation method. Under 980 nm laser exposure, when two laser fluences were applied via low (25 mW/cm^2^) and high (720 mW/cm^2^) power densities, these developed SPN showed significantly better NIR-II brightness compared to the organic NIR-II fluorescent agents, owing to their excellent stability and several other parameters such as high quantum yield etc. The improved biocompatibility along with high NIR-II fluorescence allowed them to utilize this for a real-time, whole-body visualization of glial vessels along with cerebral ischemic stroke detection in tumors with extra clarity. The significant PTT effects and NIR-II imaging capability of these NPs were realized on 4T1 tumor bearing mouse model (Figure 5) [61].

Colorectal cancers (CRC) are often difficult to treat due to their relapsing nature and cannot be completely eliminated either by surgery or by chemotherapy. Therefore, theranostic NPs having imaging and photo sensitivity would be useful, particularly for the detection and treatment of disseminated small nodules. McCarthy et al. (2021) developed a tumoroid technology for the clear understanding of NPs interaction with the 3D tumor micro-environment. CD44 targeting hyaluronic acid (HA) coated hybrid D-A polymer particles (HDAPPs) was developed to demonstrate the proof of this concept. These hybrid NPs were composed of photosensitive polymer, poly[4,4-bis(2-ethylhexyl)-cyclopenta[2,1-*b*;3,4-*b*’]dithiophene-2,6-diyl-*alt*-2,1,3-benzoselenadiazole-4,7-diyl] (PCPDTBSe), functionalized with HA, to form HA-BSe NPs, and evaluated in 3D. Monitoring of NPs transport in 3D organoids showed uniform diffusion of non-targeted HDAPPs in comparison to attenuated diffusion of HA-HDAPPs due to the nanoparticle-matrix interactions. Computational diffusion analysis has provided more information, and thereby proved that HA-HDAPPs transport was due to the diffusion. On laser irradiation, only HA-BSe NPs were capable of substantially enhanced the tumoroid toxicity. Nevertheless, the restricted entry of CD44-mediated theranostic NPs in the tumoroid, their targeting localization, and enhanced PTT in 3D tumeroid might be beneficial for understating a more complex tumor micro-environment in-vivo [79].

Another critical problem in translating theranostic nanomaterials into clinical side is associated with its severe side effects which are due to the long-term accumulation inside the body. It has been therefore, a great challenge to integrate non-accumulated characteristics, diagnosis, therapeutic functions under single room of nanomedicine. Specifically, NIR responsive functional materials with deep penetration and low scattering properties are hardly explored for developing novel NIR sensitive hybrid materials. One of the most dangerous issues related to novel nanomaterials is the bio-safety in terms of accumulation in the body when administered orally or parenterally [80,81,82,83,84,85,86]. The 2 nm sized PPy-based functional NPs were made by single step green technique, having fluorescence (FL)/PAI/NIR II trimodal imaging. These photostable NPs had 33.35% PTCE at 808 nm and was increased to 41.97% at 1064 nm, respectively. The well-designed ultra-small PPy-PEG NPs showed an improved tumor homing function along with better renal clearance. These highly biocompatible, photo theranostic NPs had excellent in-vitro and in-vivo results for NIR-II-imaging guided PTT effectively [51].

Various SPN based multifunctional theranostic nanomaterials with PAI guided PTT therapeutic capability have been studied for noninvasive mode of cancer detection and tumor ablation. To achieve such desired properties, it is mandatory to have excellent optical properties such as significantly high absorption along with PTCE. Chang et al. (2019) designed theranostic Pdots, which were composed of BDT (Benzodithiophenedione-based polymer), made by a simple nano-precipitation method. The Pdots had high cytocompatibility, along with excellent stability with improved optical properties. The in-vitro/in-vivo experiments confirmed its potential application as dual functional PAI/PTT using MCF-7 cells and tumor model. Even a low dose of this nanomaterials enabled excellent therapeutic benefits on in-vivo model (MCF-7 cancer cell bearing tumor model [87].

Photo-sensitive theranostic NPs in the NIR-II window (1000–1700 nm) have been emerging as an excellent platform for personalized medicine mainly due to their low cytotoxicity and high tissue permeability via non-invasive mode. There have been various such NPs for multi-purpose. The development of metabolizable NIR-II nanoagents for imaging-guided treatment are of great importance for noninvasive diagnosis of pathological conditions such as tumors and eradicating them effectively. To realize such goal, Men et al. (2020) developed metabolizable and highly NIR-II absorbing Pdots, for the first time, for PAI guided PTT. The ultrasmall (4 nm) sized Pdots were composed of D-A π-conjugated polymer (DPP-BTzTD) and were made using nano-reprecipitation method. These pdots showed good biocompatibility, significant photostability, bright photoacoustic signals along with high 53% PTCE. Intravenously injected Pdots showed excellent PTT under a 1064 nm laser irradiation condition with 0.5 W cm^−2^, on 4T1 tumor-bearing mice model and showed a rapid clearance from the body. The pilot study paves a way and clearly indicates their efficacy for future clinical experiments [88].

A photo-chemo approach by SPN based theranostic system was studied by Chen et al. (2019), where they have combined diketopyrrolopyrrole-based SP and polystyrene-*b*-poly(*N*-isopropyl acrylamide-*co*-acrylic acid) (PDPP3T@PSNiAA NPs) as pH/NIR light-sensitive DOX release in-vitro and in-vivo. This strategy was achieved by well-designed fabrication of photo/pH-responsive PSNiAAx by RAFT polymerization method. Later, PSNiAA was modified with PDPP3T in order to simultaneously achieve both PTT and pH/thermo-sensitive DOX release in single entity. The as-made 70nm sized NPs showed very high PTCE (η = 34.1%) and excellent photoacoustic (PA) brightness. The in-vivo analysis on HeLa tumor nearing mouse model confirmed its potential benefits for photo-chemotherapy [55].

### 4.2. PDT

For excellent PDT benefits on various cancers, it is important to have photosensitizers with high photo-induced ROS generation efficiency, high biocompatibility with significant photo toxicity. To achieve this, a D-A CPs of dibenzothiophene-S,S-dioxide were made as effective photosensitizing agents (PTAs) for PDT. The resulting copolymer PTA5 (4-octyl-*N*,*N*-bis(4-(4,4,5,5-tetramethyl-1,3,2-dioxaborolan-2-yl)phenyl)aniline with 3,9-dibromo-11,11-dioctyl-11*H*-benzo[*b*]fluoreno[2,3-*d*]thiophene-5,5-dioxide) showed strong green light emission with high photoluminescence quantum yields owing to the intercrossed excited state of local existed and charge transfer states. Polymer Pluronic F127 matrix was used to encapsulate these active moieties and showed 79–91 nm with negative surface charge (−35.7 to −27.3 mV). Under 800 nm excitation, those developed NPs had large two-photon absorption cross section of 3.29 × 10^6^ GM along with good aqueous photostability and was determined to be good in bio-imaging as well. PTA5 NPs had very high tissue-permeability up to 170 μm for hepatic vessels and 380 μm for blood vessels as confirmed on mouse ear. In addition, the developed PTA5 NPs exerted high ROS generation property upon laser irradiation. PTA5 NPs showed excellent tumor inhibition effects under 400–700 nm light irradiation at 50 mW cm^–2^/5 min every other day. These findings clearly showed that PTA5 NPs can be utilized as excellent PTAs for PDT [47].

It is a well-known fact that PDT effects can be hindered by hypoxia in tumors [89]. In order to overcome this limitation, SPN nanoprodrug (SPNpd) have been made which generate ROS and imparted hypoxia sensitized chemotherapy. These self-assembled prodrug NPs were composed of grafted PEG conjugated with a chemo agent, via hypoxia-sensitive linkers. The 30 nm sized NPs showed tumor localization in 4T1 xenograft model, where it exerted PDT/chemo effects synergistically, inhibiting the tumor growth. This study showed the first hypoxia-activatable phototherapeutic polymeric prodrug system with a high potential for cancer treatments [90].

The therapeutic outcome in PDT is greatly influenced by the structural and functional property of photosensitizers used. Tang et al. (2017) developed Chlorin e6 (Ce6) doped with photo-cross-linkable Pdots for PDT. This smart engineered Pdots had side chains of oxetane groups specifically for photo-controllability. The leaching of entrapped Ce6 molecules can be avoided greatly because of the photo-cross-linking reaction by which it forms an interpenetrated structure with SPN moiety of Pdots. The in-vitro therapeutic effects on gastric adenocarcinoma cells showed significantly better PDT effects via ROS generation under low dose of light irradiation (∼60 J/cm^2^). The in-vivo therapeutic analysis on human gastric cancer cell (SGC-7901) tumor-bearing nude mice (Figure 6) proved their efficacy and could be beneficial for treating many such solid tumors. It was apparently proved that photo-cross-linkable Pdots doped with photosensitizer could be excellent candidates for PDT [91].

The ROS responsive SPN-based pro-therapeutic agents were engineered through covalent modification of SPNs with caged therapeutic agents via hypoxia- or ^1^O_2_-cleavable linkers. When photo-irradiated, SPNs utilize oxygen to produce singlet oxygen, enabling PDT, while breaking hypoxia- or ROS-cleavable linkers to have on-demand drug release and in-situ remote activation of pro-therapeutic agents. Such remote activation of SPN-based pro-therapeutic agents would be beneficial for inducing DNA damage, RNA degradation, protein biosynthesis crippling, or activation of the immune system in tumors. Integration of such strategies, where PDT and NIR synergistically eradicate tumors, and metastases without even a chance for relapse [92]. PAI guided PDT organic agents toward lysosome-targeting is of great challenge, though they are highly effective incase developed strategically. Lysosome-targeting boron-dipyrromethene (BODIPY) NPs were designed by loading NIR absorbed BODIPY dye within amphiphilic DSPE-mPEG5000 for lysosomal PAI and acid-sensitive PDT against cancer cells under NIR light [93].

Real-time intra tumoral molecular O_2_ detection by SPN is important in early cancer diagnosis. Conversely, PDT could be achieved by super toxic ^1^O_2_ produced on site, with O_2_ sensing, and is a very significant cancer therapeutic strategy. To do so, negatively charged iridium (III) complex-hyperbranched phosphorescent CP dots for hypoxia imaging and effective PDT were systematically developed. The incomplete energy transfer between the polyfluorene and the iridium (III) complexes enabled ratiometric accurate O_2_ sensing. Furthermore, the O_2_-dependent emission lifetimes were also used in photoluminescence lifetime imaging and time-gated luminescence one for eliminating the autofluorescence remarkably to enhance the signal-to-noise ratio of imaging. Interestingly, the designed Pdots were able to generate toxic ^1^O_2_ efficiently in aqueous media. Image-guided PDT on cancer cells was studied in detail by confocal laser scanning microscope. To the best of our knowledge, it represents the first example of the negatively charged conjugated polymer dots hyperbranched with the cored iridium(III) complex for both hypoxia imaging and PDT of cancer cells simultaneously [94].

The cell penetrating peptide (CPP) modified SPN Pdots were doped with a photosensitizer for PDT applications. The as-made SPN dots showed excellent ^1^O_2_ production. Both in-vitro and in-vivo analysis confirmed that the CPP functionalized SPN Pdots possessed high cellular uptake which in turn improved the anti-cancer efficacy. Such novel CPP modified Pdots loaded photosensitizer theranostic systems hold great opportunities in treating various cancers. In-vitro efficacy was assessed using SGC-7901 cells (Human gastric cancer cell line) which was translated in to an in-vivo tumor model [95].

The degree of intracellular O_2_ level or hypoxia has been considered as an early indicator of cancers, and many efforts have been made to develop responsive drug delivery systems and SPN based NPs for targeting such hypoxic environments in cancers [96,97,98,99,100,101,102]. PDT agents can typically make use of such low oxygen environments to generate ROS, which in turn effectively eradicate cancerous tissues and cells. An early diagnosis and therapeutic platform based on phosphorescent Pdots having Pt(II) porphyrin as an oxygen-sensitive phosphorescent group and ^1^O_2_ photosensitizer was developed. The as-made Pdots were able to detect O_2_ levels, and the results showed that HepG2 cells when incubated with Pdots showed longer lifetimes under hypoxia, and time-resolved luminescence images showed a higher signal-to-noise ratio after gating off the short-lived background fluorescence. Quantification of O_2_ is realized by the ratiometric emission intensity of phosphorescence/fluorescence and the lifetime of phosphorescence. As such, these Pdots demonstrated excellent PDT effects in-vitro. The major limitation in this study was the lack of further investigation on toxicity and efficacy using animal model [103].

### 4.3. Combined PTT/PDT

Integrated theranostic nanoplatforms for targeted PTT/PDT strategy is having high relevance in the medical arena but is still challenging. Recently a “sense-and-treat” regimen based on SPNs was developed for ratiometric bioimaging of phospholipase D (PLD) activity and combined PTT/PDT. Thus, the developed PSBTBT NPs served not only as PPT agents but also as fluorescent quenchers of Rhodamine B (Rhod B) through a PLD-cleavable linker. Ce6 was used as a PDT agent. The obtained nanoplatform (PSBTBT-Ce6@Rhod NPs) showed high PDT/PTT performance upon single laser irradiation. The PTT/PDT combined therapy achieved more efficient tumor inhibition results as compared with the single treatments. In addition, the overexpressed biomarker PLD in tumor tissue will cleave Rhod, leading to the fluorescence recovery of Rhod B and thus allowing the activatable fluorescence imaging of tumor and targeted phototherapy [75].

In another report, SPN was used for PAI-guided combined PTT/PDT. Authors made triplet tellurophene-based SP (PNDI-2T) with efficient tin-free direct heteroarylation polycondensation. The PNDI-2T NPs displayed substantial NIR absorption with high cyto compatibility along with an enhanced ROS generation, high PTCE ~ 45% and a high ROS yield (ΦΔ = 38.7%) when exposed under 808 nm laser irradiation. 4T1 tumor model was used to confirm their efficacy and proved that these well-made NPs could be a potential PAI-guided PTT/PDT agents for cancer theranostics (Figure 7). This study provides a new route to developing highly efficient and low cytotoxic agents for PAI-guided PTT/PDT [104].

Biomimetic phototheranostic nano-agents that are capable for targeting cancer-specific fibroblasts in the tumor microenvironment are very promising candidates, especially for personalized anti-cancer medicines. Though such theranostic agents have various applications, their efficacy could be hindered by various tumor micro environmental factors such as hypoxia. Herein, organic multimodal PTT nanoagents for the improved multimodal imaging-guided has been developed for cancer theranostic purpose. These NPs contains NIR absorbing SPN further covered with the cell membranes of activated fibroblasts (AF) to finally have AF-SPN for selectively targeting cancer-specific fibroblasts, allowing better tumor localization than the un-coated one after systemic administration in living mice. AF-SPN were able to produce enough PTT/PDT effects along with their dual imaging capability via PAI and NIR fluorescence. The in-vitro experimental results were further extended to in-vivo 4T1 bearing tumor model [52].

SPs with high photostability, strong NIR absorptivity, and efficient PTCE are highly desired to be considered as an ideal photo theranostic agent. Here, a heavy atom free tri-component SPs of PTVT has been made through a Stille coupling reaction. PTVT showed high singlet oxygen quantum yield (^1^O_2_ QY) ~42.2% in DCM and the ^1^O_2_ generation ability for the DSPE-PEG coated PTVT NPs remained higher. These multifunctional NPs were made through nanoprecipitation technique. Owing to the PDT and PTT synergistic therapy (with PTCE ~ 52.6%), PTVT NPs had low IC50 of 5.9 μg/mL on A549 cell-line under NIR exposure. The dark toxicity of such NPs was minimal even at high concentrations. In in-vivo study, a similar pattern was observed with exceptionally high PTT and PDT efficacies on living mice model with a good organ compatibility for the NPs as confirmed by histopathology [105].

Even though most of the organic SPN are having excellent characteristics for phototherapy, their weak emission spectra, mainly due to the aggregation-caused quenching phenomenon (ACQ), have been limiting their bio-imaging capability. D–A–D type compound namely BODIPY-TPA was designed by conjugating triphenylamine (TPA) with a BODIPY structure. PEG functionalization was given to BODIPY core in order to enhance the bio-safety of the developed SPN. Through self-assembly mechanism, BODIPY-TPA could be formed as NPS (BODIPY-TPA NPs) and it showed remarkable NIR fluorescence due to its AIE mechanism, owing to the twisted structure of TPA. Additionally, BODIPY-TPA NPs generated ^1^O_2_ and heat simultaneously on a single laser (635 nm) irradiation. The PTCE was found to be 20.7%. The lysosome targeting property was confirmed on A549 cell-line, in-vitro and it had IC50 ~ 28.45 μg/mL. These preliminary studies would open up new ventures for BODIPY based NPs for bioimaging guided combined PDT/PTT synergistic cancer treatment [54].

High ROS generation and PTCE are the key requirements for emerging therapeutic materials for combined PDT/PTT therapy. Conversely, organic nanomaterials showed poor photostability in aqueous media due to ACQ, and could negatively affect their bio-imaging applications. To troubleshoot this, D–A–D fashioned organic small molecule (T-BDP) NPs were made of BODIPY and TPA meticulously. Moreover, electron-withdrawing 1,8-naphthalenediimide (NI) was functionalized onto the BODIPY core to enhance intramolecular charge transfer, enabling a red shift towards the NIR window. T-BDP showed significantly high AIE performance, owing to the twisted TPA attached on the BODIPY. T-BDP NPs showed high NIR emission in water when exposed to mono laser treatment at 635 nm, which in turn result in the simultaneous generation of ROS and heat. The PTCE of T-BDP NPs was determined to be 50.9%. The low dark toxicity and high photocytotoxicity of T-BDP NPs were confirmed on A549 cells using the MTT and the AM/PI staining method. Due to the strong emission of T-BDP NPs**,** their accumulation and subcellular localization in cancer cells were observed using a laser confocal fluorescence microscope. The results demonstrated that T-BDP NPs were mainly located in the lysosomes of cancer cells. Thus, the as-prepared small molecule-based AIE nanoparticles hold great potential for fluorescence imaging-guided PDT/PTT synergistic tumor therapy [106].

Phototherapeutic limitations, such as low light penetration depth and insufficiency of photothermal agents, often hamper the efficacy of PDT and PTT. To solve these, Peroxynitrite (ONOO−), an oxidizing and nitrating agent involved in various physiological and pathological processes, has been generated in-situ by SPN. For this, a cyanine dye-based (Cy7) SPN was developed for improved phototherapy by in situ generation of ONOO−. The Cy7 units in the SPN have dual functions as photosensitizer to produce ROS for PDT, and heat source for PTT by NO gas release from N-nitrosated napthalimide (NORM) at the same time. Since NO can react quickly with superoxide anion to generate ONOO−, the enhanced phototherapy could be achieved by in-situ ONOO− produced by PCy7-NO under NIR conditions [107].

Multifunctional theranostic systems having imaging and theranostic properties are of great interest especially in clinical applications. For example, a mono-laser activatable lipid-micelles modified SPN dots and a photosensitizer (Pdots/Ce6@lipid–Gd–DOTA micelles) for combined MRI/PAI and PDT/PTT bi-modal strategy have been developed for therapeutic benefits on cancer. The aqueous dispersible SPN micelles were composed of poly[2,6-(4,4-bis-(2-ethylhexyl)-4*H*-cyclopenta[2,1-*b*;3,4-*b*′]-dithiophene)-*alt*-4,7-(2,1,3-benzo-thiadiazole)] dots (Pdots) core having Ce6 molecules within them, followed by lipid–PEG outlayer functionalized with gadolinium-1,4,7,10-tetraacetic acid. The prepared Pdots/Ce6@lipid–Gd–DOTA micelles had excellent cytocompatibility, with significantly relevant MRI/PAI capability, enabling simultaneous structural as well as topological information concerning malignancies. The loaded Pdots and Ce6 within the micelles had high NIR absorption at 670 nm enabling simultaneous PTT/PDT to obtain improved synergistic anti-cancer effects both in-vitro and in-vivo. A HepG_2_ bearing tumor model was used to confirm the efficacy of the prepared Pdots/Ce6@lipid–Gd–DOTA micelles [108].

### 4.4. Photo-Immuno Therapy

One of the great advantages of phototherapy is that it can effectively kill tumor cells while inducing immunogenic cell death (ICD) to kick off a systemic antitumor immune responses by redistributing and activating immune effector cells, cytokines and memory T lymphocytes transformation [109]. Both PTT and PDT therapeutic strategies have been combined with immunotherapy for better efficacy [110]. Even though there have been various NPs based on inorganic/inorganic and their composites [111], little is known for SPN based systems. Li et al. (2021), developed a SP nanoadjuvant (SPN_II_R) for photo-activable drug release for NIR-II/ PTT/immunotherapy. The nano adjuvants were made using hydration-sonication method. SPN_II_R consisted of a SPN core as an NIR-II photothermal converter, which is doped with a toll-like receptor (TLR) agonist as an immunotherapy adjuvant and coated with a thermally responsive lipid shell. Upon NIR-II photoirradiation, SPN_II_R effectively generated not only thermal effects enabling ICD, but also the melting of lipid layers for effective on-demand release of the TLR agonist. It was, therefore, concluded that the combination of ICD and activation of TLR7/TLR8 could enhance the maturation of dendritic cells, resulting in the amplification of anti-tumor immune responses [72].

One of the most relevant goal of combining nanomedicine with immunotherapy is to have patient compliance and safety during treatment period with excellent efficacy. However, one of the main challenges in immunotherapy is its difficulty in controlling immune response with spatiotemporal precision. To overcome such limitations, photo-sensitive activatable polymeric pro-nanoagonist (APNA) have been reported. APNA was selectively controlled by NIR-II waves for combined photo immunotherapy. These smart NIR probes were made by covalent conjugation of an immunostimulant onto semiconducting transducer via heat sensitive linker molecule. Interestingly, when the NIR-II sensitive probes were exposed under NIR-II laser, it had not just PTT tumor ablation and immunogenic cell death, but additionally the heat sensitive linker molecules broke to un-cage the agonist for in-situ immune activation in 8 mm seated solid 4T1 mouse tumor as well. This kind of well-regulated combined PTT/in-situ immunotherapy could have great potential for treating highly metastatic cancers (Figure 8) [76].

Tumor immunometabolism contributes substantially to tumor proliferation and immune cell activity, and thus plays a crucial role in the efficacy of cancer immunotherapy. Modulation of immunometabolism to boost cancer immunotherapy is mostly based on small-molecule inhibitors, which often encounter the issues of off-target adverse effects, drug resistance, and unsustainable response. In contrast, enzymatic therapeutics can potentially bypass these limitations, but have been less exploited. Herein, an organic polymer nanoenzyme (SPNK) with NIR photoactivatable immunotherapeutic effects was reported for photodynamic immunometabolic therapy. SPNK was composed of a SP core conjugated with kynureninase (KYNase) via PEGylated singlet oxygen (^1^O_2_) cleavable linker. Upon NIR photoirradiation, SPNK generates ^1^O_2_ not only to exert photodynamic effect to induce the immunogenic cell death of cancer, but also to unleash KYNase and trigger its activity to degrade the immunosuppressive kynurenine (Kyn). Such a combinational effect mediated by SPNK promoted the proliferation and infiltration of effector T cells, enhancing systemic antitumor T cell immunity, and ultimately permitting inhibition of both primary and distant tumors in living mice. Therefore, this study provided a promising photodynamic approach toward remotely controlled enzymatic immunomodulation for improved anticancer therapy [112].

### 4.5. Photo-Radio Therapy

Photo-radio therapy typically combines principles of phototherapy along with radiation therapy and such combinatorial therapy is expected to improve the clinical outcome than that of their individual therapies [113,114,115,116,117,118,119,120].

For example, radio therapy suffers from tumor hypoxia [121,122,123,124], and in such cases, it is important to supply sufficient O_2_ in the tumoral area to enhance the growth of tumor cells, as they are in need of metabolizing O_2_ [125]. These limitations can be avoided by hybrid semiconducting organosilica-based O_2_ nanoeconomizer pHPFON-NO/O_2_ platform. There are two major mechanisms; (1) When pHPFON-NO/O_2_ interacts with the acidic tumor microenvironment, releases sufficient NO for effective O_2_ conservation endogenously; (2) O_2_ generation was in response to mild PTT effect for O_2_ infusion exogenously. Additionally, PTT effect can be increased for ablating residual tumor cells with radio-characteristic nature of NPs. This “reducing expenditure of O_2_ and broadening sources” technique substantially enhance significantly reduce tumor hypoxia through many ways, improving the therapeutic efficacy in-vitro and in-vivo. The preliminary experimental results clearly demonstrated the synergy between on-demand thermo-regulated PTT and oxygen-elevated radiotherapy for complete tumor eradication (Figure 9) [126].

## 5. Future Perspectives

Even though there has been tremendous progress in SPN based theranostic drug delivery systems, it is hard to believe that most of the work stopped just as “published article”. Therefore, one has to critically think of bringing them from lab to market. In most cases, the in-vivo results proved that SPN are nontoxic to major organs, such as the liver, kidney, heart, brain, stomach, and intestine, owing to their neutral character. However, novel and complex SPN systems have to be carefully evaluated for their long-term toxicity. In many cases, the novel SPN based work limited to in-vitro experiments making it difficult to understand their applicability for clinical purposes.

In the majority of the experimental results, in-vitro results look promising, however, clinical relevance was not discussed or studied, and this must be taken into account. Wide variety of semi conducting polymers have been synthesized not just for PTT/PDT applications, but for imaging guided phototherapy as well, showing the huge progress in organic synthesis. However, chemists need to understand the urge of in-depth testing of such novel organic/inorganic hybrid materials in detail prior to biological applications. Just in-vitro analysis would not be sufficient in most of cases, and in those situations, in-vivo assessment of novel SPN based photo agents should be tested vigorously for its long term exposure to human body and the environment.

It is a well-known fact that ultra-small sized NPs have easy renal clearance from the body, and it is very important to reduce the long-term accumulation of novel NPs inside the human body. One has to critically think of reducing particle size to ultra-small nano range, typically less than 10 nm. While there are few such reports, as discussed in our article, at the same time, many SPN have larger particle size (>10 nm to 200 nm) and their in-vivo fate in terms of body accumulation and clearance is still questionable. Typically, an ideal SPN system should possess good anti-cancer therapeutic efficacy without damaging the neighboring or surrounding cells with an easy body clearance. The selectivity of SPN is also an issue on which researchers should give more attention for further improving the therapeutic outcome. 

Nano-precipitation was reported to be the most well used method for synthesizing NPs of semiconducting polymers, however, the obtained NPs, had size in the range of 10–200 nm depending up on the type of material as shown in Table 1. One has to think of utilizing other methods that could produce better size in the ultra-small range, which might facilitate easy renal clearance, reducing the risk of long-term accumulation in the body as mentioned before.

In addition, scientists have to think of making composites or hybrids of SPN with inorganic clay materials which might be useful for improving not only the efficacy but also enhancing bio-availability with reduced organ toxicity. Clay materials, such as layered double hydroxide (LDH), montmorillonite (MMT), and halloysite NPs, possess great functionality owing to their easy modification properties and intercalating nature into the interlayers. Typically, suitable drug molecules can be stabilized in the interlayer space of such clay NPs so that one can achieve pH responsive drug delivery during SPN assisted phototherapy. These kinds of approaches would protect the integrity of loaded therapeutic agents until they reached the targeted tumor sites. 

## 6. Conclusions

The semi conducting polymer based theranostic nanomaterials are expected to hold great future in phototherapy, especially for PTT and PDT as mono and combined therapy, owing to their exceptionally good opto-electronic properties. There have been tremendous progress in developing new materials for phototherapy along with immuno/radio therapies. However, the major issues such as long-term toxicity of SPN based theranostic NPs remain unresolved. It is certain that SPN based research will keep growing, however scientists have to focus on their negative side as well, making them highly suitable for actual applications in cancer therapy. We believe that future studies on their safety, long term accumulation and clearance etc. would be highly beneficial for bringing SPN from bench to market.

## Figures and Tables

**Figure 1 polymers-13-00981-f001:**
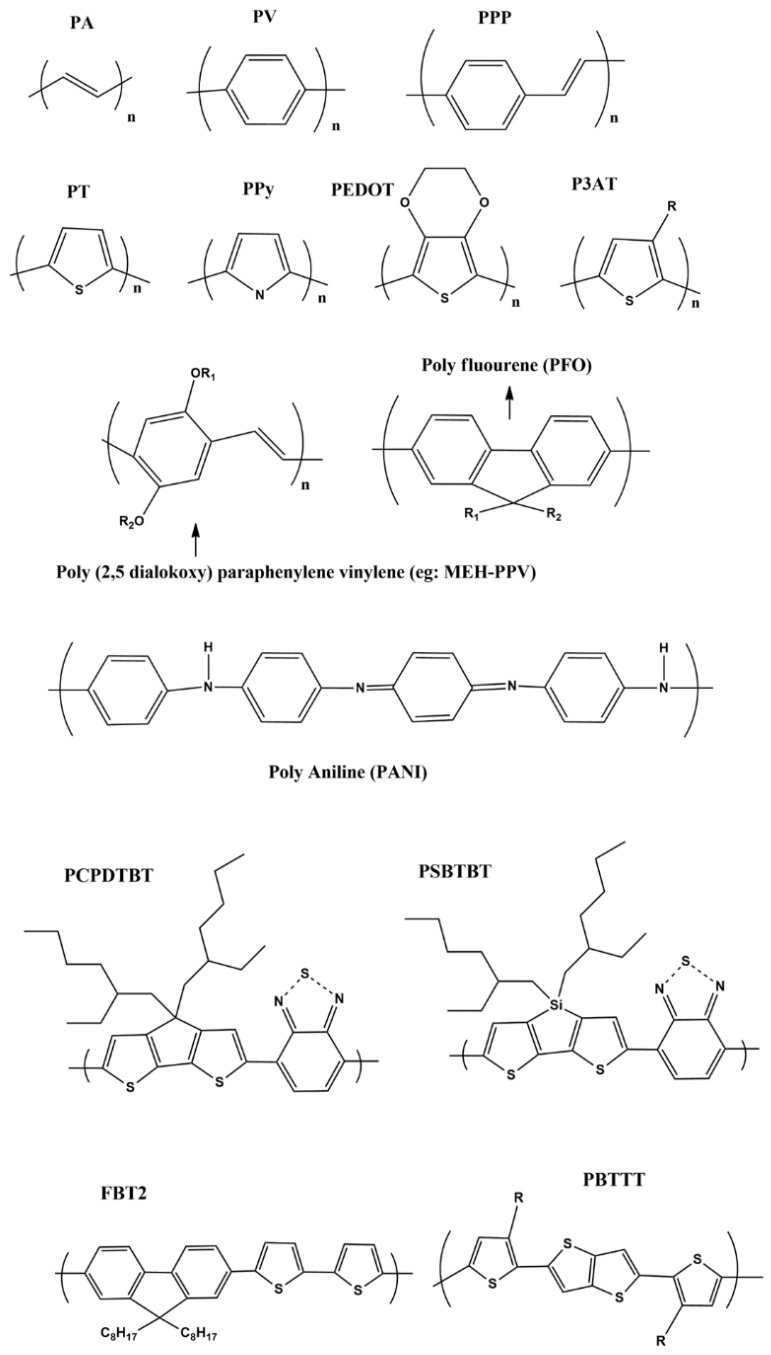
Shows various semi conducting polymers from first to third generations: PA-poly acetylene; PV-poly paraphenylene vinylene; PPP-poly para phenylene; PT-poly thiophene; PPy-poly pryrrole; PEDOT-poly ethylene dioxythiophene; P3AT-poly 3(alkyl) thiphene (R-Alkyl group); PCPDBT-poly(2,6-[4,4-bis-(2-ethylhexyl) 4*H*-cyclopenta (2,1-*b*;3,4-*b*^1^)dithiophene-*alt*-4,7-(2,1,3-benzothiadiazole)]; PSBTBT-Poly [(4,4′-bis(2-ethylhexyl)dithieno[3,2-*b*:2′,3′-*d*]silole)-2,6-diyl-alt-(2,1,3-benzothiadiazole)-4,6-diyl]; FBT2-poly(9,9-dioctylflourene-*co*-bithiophene); PBTTT-poly(2,5-bis(3-alkylthiophen-2-yl)thieno[3,2-*b*]thiophene respectively.

**Figure 2 polymers-13-00981-f002:**
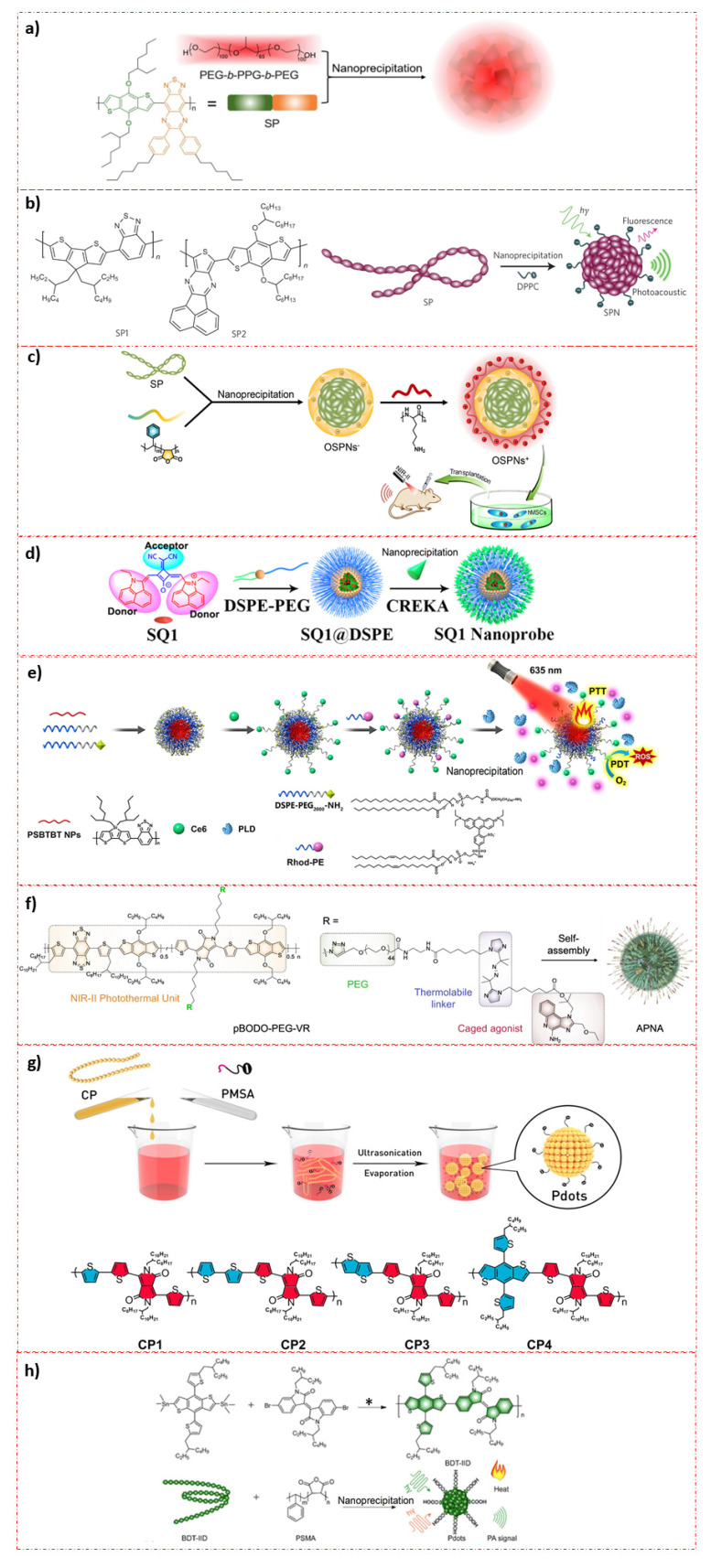
Preparative techniques for various SPN based photo-theranostic materials: (**a**) Illustration of the SPNs preparation from SP and PEG-*b*-PPG-*b*-PEG using nanoprecipitation method. (Reprinted Permission from American Chemical Society, 2020) [46]; (**b**) Molecular structures of SP1 and SP2 used for the preparation of SPN1 and SPN2, respectively. SPNs made through nanoprecipitation. SP is represented as a long chain of chromophore units (red oval beads). DPPC contains a short hydrophobic tail and a charged head and is illustrated as a string with a dark green ball at its end. (Reprinted permission from Nature Nanotechnology, 2014) [69]; (**c**) illustration of the preparation procedure of OSPNs^+^ and the photoacoustic labeling of hMSCs after transplantation. (Reprinted permission from American Chemical Society, 2018) [70]; (**d**) Molecular engineering and nano functionalization of Squaraine dye SQ1 for NIR-II/PA Bimodal Imaging and Photo-thermal ablation of metastatic breast cancer. (Reprinted permission from American Chemical Society, 2020) [74]; (**e**) Schematic Illustration of PLD-Activatable Tumor Image and PTT/PDT Combined Therapy (Reprinted permission from American Chemical Society, 2021) [75]; (**f**) Chemical structure of pBODO-PEG-VR and preparation of APNA (Reprinted permission from Nature, 2021) [76]; (**g**) Schematic illustration of preparation for Pdots (Reprinted permission from American Chemical Society, 2016) [48]; (**h**) Synthetic route of conjugated polymer BDT-IID (*****) Pd(PPh3)4 and toluene, 110 °C and preparation of BDT-IID Pdots for PAI-guided PTT (Reprinted permission from American Chemical Society, 2018) [49].

**Figure 3 polymers-13-00981-f003:**
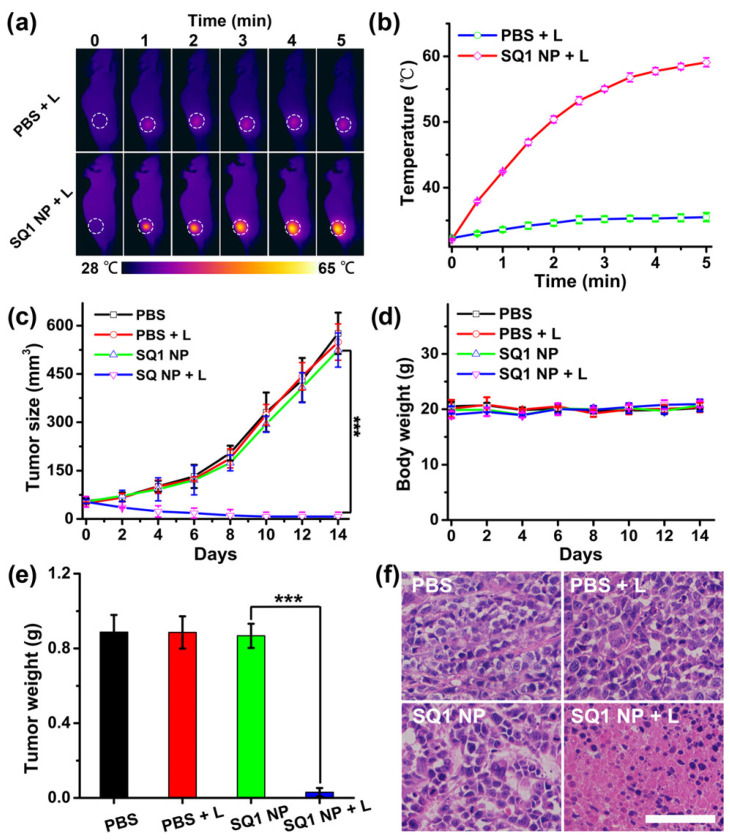
PTT effects of Squarine dye based copolymer NPs on breast tumor burden on s mice. (**a**) Thermal imaging; (**b**) Thermal variations in tumor during PTT; (**c**) Tumor growth inhibition and (**d**) body weight analysis during treatment period; (**e**) Weight measures of tumor lesions post treatment with PBS, PBS + L, SQ1 NP, SQ1 NP + L. L refers NIR laser (915 nm, 0.5 W/cm^2^). Data present as mean ± SD, *n* = 5 (*** *p* < 0.01). (**f**) Histochemical analysis of tumor sections treated with PBS, PBS + L, SQ1 NP, SQ1 NP + L. The scale bar is 50 μm. (Reprinted permission American Chemical Society, 2020) [74].

**Figure 4 polymers-13-00981-f004:**
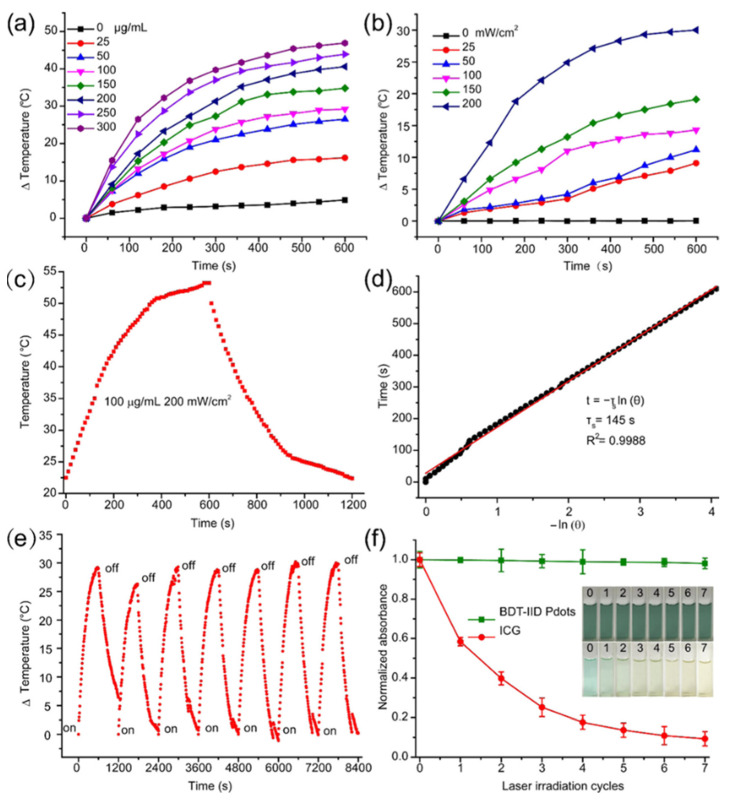
PTT characteristics of BDT-IID Pdots. PTT heating curves of the BDT-IID Pdots (**a**) with different concentrations upon 200 mW/cm^2^ laser exposure at 660 nm and (**b**) with different laser power densities at 100 μg/mL. (**c**) Photothermal effect of the BDT-IID Pdots dispersions under laser irradiation at 660 nm (200 mW/cm^2^). Irradiation terminated after 600 s. (**d**) Time constant for heat transfer was determined to be τs = 145 s by applying the linear time data from the cooling period (after 600 s) versus negative natural logarithm of driving force temperature, obtained from the cooling stage of (**c**); (**e**) thermal variations of the Pdots under laser exposure at 200 mW/cm^2^ for seven light on/off cycles (10 min of irradiation for each cycle). (**f**) Change in absorbance intensity of BDT-IID Pdots and ICG after repeated laser irradiation (*n* = 7). The figure inserts show the changes of BDT-IID Pdots and ICG after repeated laser irradiation (*n* = 7). (Reprinted Permission American Chemical Society 2018) [49].

**Figure 5 polymers-13-00981-f005:**
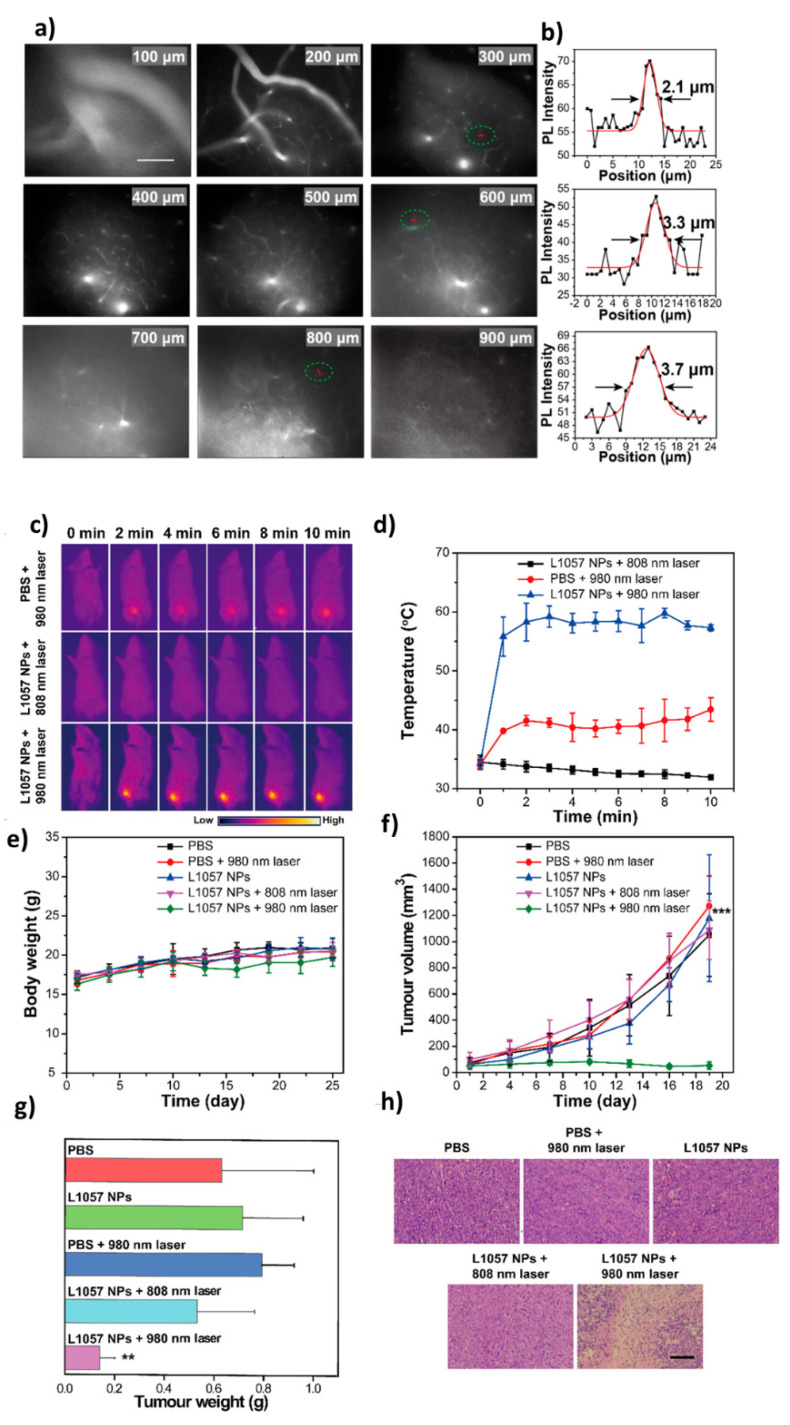
Real-time in-vivo NIR-II fluorescence microscopic imaging of mouse brain vasculature. (**a**) Cerebrovascular imaging at various depths (100–900 μm) after the intravenous injection of L1057 NPs. The excitation wavelength was 980 nm. Scale bar: 100 μm. (**b**) Cross-sectional fluorescence intensity profiles (and Gaussian fits (red) with fwhm indicated by arrows) along the red lines circled with green in panel a; PTT efficacy of L1057 NPs on tumors. (**c**,**d**) PTT images (**c**) and corresponding temperature changes (**d**) of 4T1-tumor-bearing mice under irradiation with an 808 (0.33 W/cm^2^) or 980 nm (0.72 W/cm2) laser. (**e**,**f**) Body weight (**e**) and tumor volume (**f**) curves of tumor-bearing mice at different time points after receiving PTT. (**g**,**h**) Tumor weight (**g**) and H&E staining (**h**) of the tumor tissues from mice sacrificed at day 18 post-PTT treatment. Scale bar: 100 μm. Results are presented as the mean ± S.D., *n* = 5. Statistical significance was calculated using one-way ANOVA with the Tukey posthoc test. *** *p* < 0.001. (Reprinted Permission from American Chemical Society, 2020) [61].

**Figure 6 polymers-13-00981-f006:**
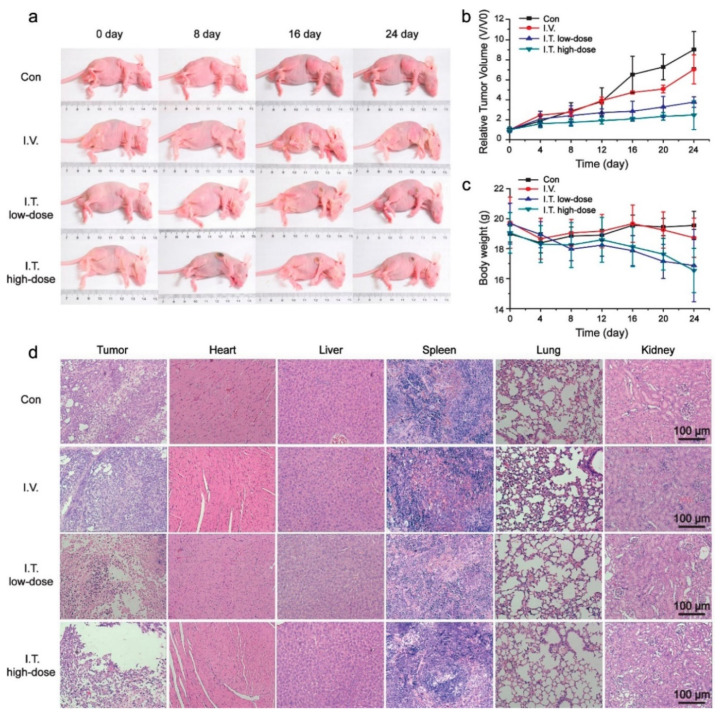
In-vivo photodynamic effect of Ce6-doped Pdots. (**a**) representative photographs for tumor-bearing nude mice during PDT. Relative tumor volume (V/V0) (**b**) and body weight (**c**) of tumor-bearing nude mice in control group (Con), intravenous injection group (I.V.), intratumoral injection low-dose group (I.T. low-dose) and intratumoral injection high dose group (I.T. high-dose). (**d**) Histopathology analysis of tumors and organs collected from tumor bearing nude mice after PDT. (Reprinted Permission American Chemical Society, 2017) [91].

**Figure 7 polymers-13-00981-f007:**
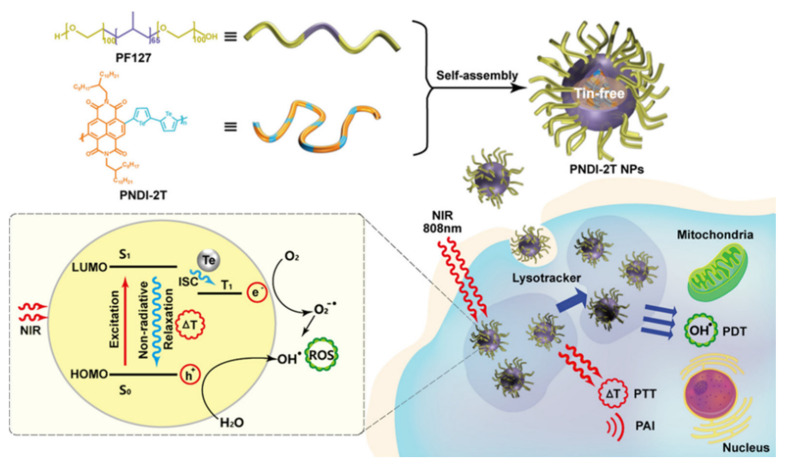
Schematic illustrations of the preparation of NPs for a PAI-Guided PDT/PTT and proposed photo-physical mechanism (Reprinted Permission American Chemical Society, 2019) [104].

**Figure 8 polymers-13-00981-f008:**
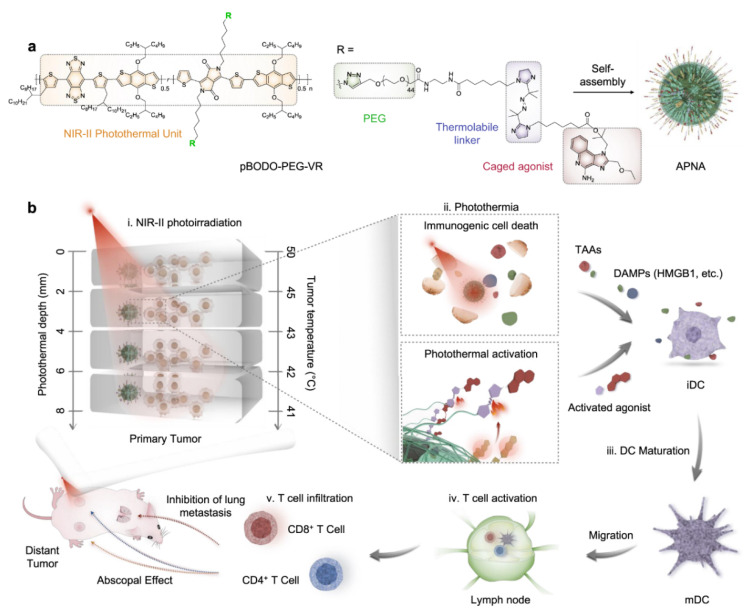
(**a**) Chemical structure of pBODO-PEG-VR and preparation of APNA. (**b**) Mechanism of antitumor immune response by APNA-mediated NIR-II photothermal immunotherapy. TAAs tumor-associated antigens, DAMPs damage-associated molecular patterns, iDC immatureDC, mDC mature DC, HMGB1 high-mobility group box 1 protein (Reprinted Permission from Nature, 2021 under creative common license) [76].

**Figure 9 polymers-13-00981-f009:**
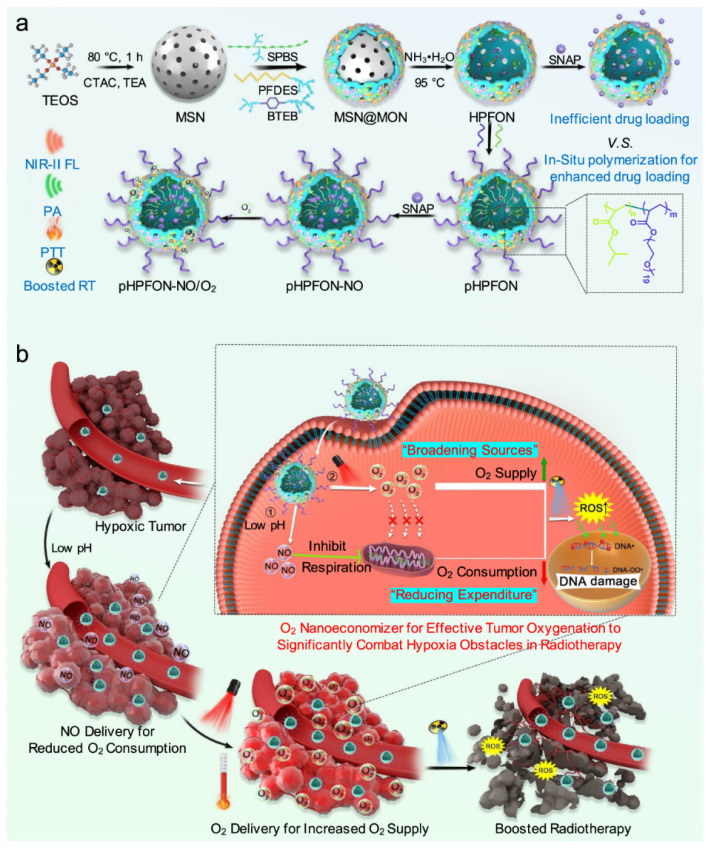
(**a**) Synthetic procedures. First, sub-50 nm SP brush/fluorocarbon/phenylene triple-hybridized HPFON prepared by deposition of bissilylated organosilica precursors onto an MSN template via hydrolysis based on the chemical homology principle and selective MSN etching through an ammonia-assisted hot water etching strategy. Then, an in situ polymerization method is applied to conjugate alkyl chains and PEG polymers onto the inner and outer shell of the HPFON for enhanced hydrophobic drug loading as well as improved biocompatibility. Finally, SNAP and O_2_ were loaded onto the resultant pHPFON to generate the pHPFON-NO/O_2_. (**b**) Schematic illustration of the binary “reducing expenditure and broadening sources” tumor oxygenation strategy by programable delivery of NO and O_2_ with pHPFON-NO/O_2_ to overcome hypoxia-associated therapy resistance for boosted anti-cancer radiotherapy. (Reprinted Permission from Nature, 2021 under creative common license) [126].

**Table 1 polymers-13-00981-t001:** Various emerging SPN systems studied during last 5 years (2015–2020).

Materials	Specification	(PTCE) *	SPN Formulation Method	Applications	Ref(s)
Diketopyrrolopyrrole polymer P(AcIIDDPP)	200 nm sized SPN with very good stability	49.5%	Stille cross-coupling reaction followed an emulsification method	Improved PTT on human epithelial cervix adenocarcinoma (HeLa) bearing mouse model	[45]
Thiadiazoloquinoxaline-based semiconducting polymer	Hydrodynamic size of SPNs is 58.9 ± 1.4 nm and the PDI is 0.35.	21.2%	Nanoprecipitation method	PTT on human brain glioblastoma cell line (U87) xenograft model	[46]
Dibenzothiophene-S,S-dioxide derivatives	High photostability, improved tissue penetration	66% **	Nano-precipitation method	Improved PDT on human epithelial cervix adenocarcinoma (HeLa) bearing mouse model	[47]
Thiophene based conjugate polymers	30 nm sized conjugated polymer	65%	Nano-precipitation method	Improved PTT effects on mouse epithelial mammary gland metastatic cancer cells (4T1) bearing tumor model	[48]
4,8-bis[5-(2-ethylhexyl)thiophen-2-yl]-2,6-bis(trimethylstannyl)benzo[1,2-b:4,5-b′]dithiophene-6,6′-dibromo-*N*,*N*′-(2-ethylhexyl)isoindigo (BDT-IID) Pdots	20 nm sized Pdots	45%	Nanoprecipitation method	PTT on Human epithelial mammary gland adenocarcinoma cell line (MCF7) bearing tumor model	[49]
Gd^3+^-PMA–PDI–PEG NPs (Gd^3+^-chelated poly(isobutylene-*alt*-maleic anhydride) (PMA) framework pendent with perylene-3,4,9,10-tetracarboxylic diimide (PDI) derivatives and poly(ethylene glycol) (PEG))	101.9 ± 2.8 nm (PMA–PDI–PEG NPs)72.6 ± 2.4 nm (Gd3^+^-PMA–PDI–PEG NPs)	40%	Nanoprecipitation method	PTT on human epithelial cervix adenocarcinoma (HeLa) tumor model	[50]
PolyPyrrole-PEG NPss	7 nm	33.35% (808 nm), 41.97% (1064 nm)	Self assembling method	Multimodal imaging and PTT on human brain glioblastoma cell line bearing mouse tumor model	[51]
poly(cyclopentadithiophene-*alt*-benzothiadiazole) (PCPDTBT)	47 nm with −20 mV	Not given	Nanoprecipitation method	Combined PTT/PDT on mouse epithelial mammary gland metastatic cancer cells (4T1) tumor bearing mouse	[52]
Poly vinylene based SPN	36 nm	Not given	Nanoprecipitation method	Improved PAI/PTT effects on mouse epithelial mammary gland metastatic cancer cells (4T1) tumor bearing model	[53]
BODIPY-TPA (Triphenylamine)	80 nm in size with −35.5 mV surface charge	20.7%	Nano-precipitation method	Improved PTT/PDT in-vitro on human epithelial lung carcinoma cell line (A549)	[54]
diketopyrrolopyrrole-based semiconducting polymer and polystyrene-*b*-poly(*N*-isopropyl acrylamide-*co*-acrylic acid) (PDPP3T@PSNiAA NPs)	70 nm sized NPs showed photo chemo effects in-vitro and in-vivo	34.1%	Co-precipitation method	High photo chemo effects on human epithelial cervix adenocarcinoma (HeLa) bearing tumor model	[55]

* PTCE-Photo thermal conversion efficiency; ** photoluminescence quantum efficiency; PDI-Polydispersity index.

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
