# Peer review of "Recent Developments on Semiconducting Polymer Nanoparticles as Smart Photo-Therapeutic Agents for Cancer Treatments—A Review"

_polymers, 2021, doi:10.3390/polym13060981_

Round 1
Reviewer 1 Report
Manuscript entitled "Recent developments on semiconducting polymer nanoparticles as smart photo-therapeutic agents for cancer treatments- a
short review" by N Sanoj Rejinold et al. is a detailed review about the semiconducting polymer nanoparticles. Authors have done a detailed review from the recent years research work.
This manuscript is good except few typo errors
Page 1: Introduction: Paragraph 2: Please abbreviate PTT and PDT (though it is abbreviated in the abstract, please abbreviate in the introduction part also)
Page 19: Paragraph 2: Line # 2 & 3 typo error (however & one)
Overall the manuscript is good and well written in detail about the future perspective also.
Author Response
"Please see the attachment."

Reviewer 2 Report
The Authors have written a review on the recent research advancements on the possible application of semiconducting polymer nanoparticles for PTT, PDT (and combined) clinic treatments of cancer.
The review, which highlights the progress (and the present limits) in this field, is well organized, interesting and readable (apart a few typos and grammatical errors).
Therefore, I suggest its publication, once the following minor revisions will be performed on the manuscript.
First of all, I think that in the introduction a basic paragraph on PTT and PDT therapies (including some introductory references) would be very useful to enlarge the possible audience of the paper.
Secondly, the paper should be re-read and checked for typos and grammatical errors.
Just to do a few examples, I list some of these errors:
1) page 1, abstract, line 7: "are published" -> "have been published"
2) page 1, introduction, line 7: "Phottherapy" -> "Phototherapy"
3) page 1, introduction, line 8: "Variety nanomaterials" -> "A variety of nanomaterials" (in several points of the text the articles are missing)
4) page 2, paragraph 2, line 1: "verities" -> "varieties"
5) page 2, paragraph 2, line 5: "mechanisms such" -> "mechanisms exist, such"
6) a reference to Table 1 should be appear in the first pages of the text
7) page 8, line 3: "has three main" -> "have three main"
8) in the first paragraph of page 11, the verb is missing in a couple of sentences
9) at page 11, the sentence "On 980 nm laser exposure, with two laser fluence were ap-plied via low (25 mW/cm2) and high (720 mW/cm2)" should be more clearly rewritten
10) page 13, line 36: "whih" -> "which"
11) page 13, line 38: "experimetnts" -> "experiments"
12) page 13, line 39: please correct "lose of dose"
13) page 13, line 41: "theranstc" -> "theranostic"
14) page 13, 5th line from the bottom: "Pdots composed" -> "Pdots were composed" (also in other points of the text "composed" is uncorrectly used)
15) page 14, par 4.2, beginning of the 2nd paragraph: "It is well known fact" -> "It is a well known fact" (and in the following line "Inorder to" -> "In order to": in a few points of the text spaces are missing)
16) page 16, first sentence of the 3rd paragraph: the verb is missing;
17) page 16, 2nd line of the 4th paragraph: "manny" -> "many"
18) page 17, last line but two: "with an exceptionally" -> "with exceptionally"
19) page 18, line 17: "could be negatively" -> "could negatively"
20) page 18, line 24: "determined" -> "was determined"
21) page 18, line 46: "composed" -> "were composed"
22) page 18, line 50: "excellemt" -> "excellent"
23) page 18, line 51: "enablng simultabeous" -> "enabling simultaneous"
24) page 19, line 10: substitute "Here" with the reference
25) page 19, line 11: "activatble" -> "activable"
26) page 19, line 20: "combinning" -> "combining"
27) page 19, line 21: "Howvevr oen" ->"However one"
28) page 19, line 29: "additioanlly" -> "additionally"
29) page 21, line 13: correct "substantially enhance significantly reduce"
30) page 23, line 4: "In majority" -> "In the majority"
31) page 23, line 13: "It is well known fact" -> "It is a well known fact"
32) page 23, line 16: "Yes" -> "While"
33) page 23, line 17: "many of SPN has" -> "many SPN have"
34) page 23, line 19: "should system possess" -> "system should possess"
Author Response
"Please see the attachment."
